# A Viable Approach to Mitigating Irreproducibility

**David Trafimow** [1,*] **, Tonghui Wang** [2] **and Cong Wang** [3]

1    Department of Psychology, New Mexico State University, Las Cruces, NM 30001, USA
2    Department of Mathematics, New Mexico State University, Las Cruces, NM 30001, USA; twang@nmsu.edu
3    Department of Mathematics, University of Nebraska Omaha Omaha, NE 68182, USA; congwang@unomaha.edu
*    Correspondence: dtrafimo@nmsu.edu; Tel.: +1-(575)-646-4023

**Abstract:** In a recent article, Trafimow suggested the usefulness of imagining an ideal universe where the only difference between original and replication experiments is the operation of randomness. This contrasts with replication in the real universe where systematicity, as well as randomness, creates differences between original and replication experiments. Although Trafimow showed (a) that the probability of replication in the ideal universe places an upper bound on the probability of replication in the real universe, and (b) how to calculate the probability of replication in the ideal universe, the conception is afflicted with an important practical problem. Too many participants are needed to render the approach palatable to most researchers. The present aim is to address this problem. Embracing skewness is an important part of the solution.

**Keywords:** replication; skewness; a priori procedure; idealized universe; precision

## 1. A Practical Way to Mitigate Irreproducibility That Often Will Work

Following the Open Science Collaboration [1] article showing an enormously high rate of replication failures, there has been much debate about whether there is a replication crisis in the soft sciences. Rather than taking a position on this issue, we commence with the obvious recommendation that whether there is a crisis or not, increasing reproducibility would be an important gain for the soft sciences. Although it is well known that a way to increase reproducibility is to increase sample sizes, this solution often is not feasible. In contrast, imagine a way to increase reproducibility without increasing sample sizes, merely by performing a procedure that is generally available, cost-free, and easy to perform. This promise may sound too good to be true, but our goal is to show otherwise. The solution depends on two preliminary issues that are necessary to understand in some detail. First, it is necessary to reconceptualize what is meant by replication. Second, it is necessary to understand the a priori procedure (APP) in some detail, along with its relevance to reproducibility. It is the juxtaposition of these two issues that renders possible the proposed solution to increasing reproducibility in the soft sciences with less than optimal sample sizes.

## 2. Rethinking Successful Replications

Traditionally, a successful replication means that an initial experiment and second experiment result in statistically significant findings in the same direction (e.g., $p < 0.05$). There are at least two alternative approaches that also use $p$-values, but in different ways. Killeen [2] proposed a transformation of the traditional $p$-value, to obtain what he considered the probability of obtaining a second finding in the same direction of the original finding. Simohnson [3] suggested an approach that focusses on the statistical power engendered by the researcher's sample size; the idea is that if a study with a larger sample was insufficient to obtain statistical significance with a reasonable probability, a study with a smaller sample size is even less trustworthy. The result of the less powerful study

is not necessarily wrong; but it is suspect if the effect cannot be detected with the more powerful study.

As Trafimow [4] explained, these approaches are problematic for a variety of reasons, but the effect size issue is sufficient. The foregoing approaches imply that the larger the effect size, the more replicable the effect; it is much easier to replicate a strong effect than a weak effect. Although this implication may seem unobjectionable, there is a fatal flaw that might be termed the Michelson and Morley (M & M) problem. In the latter part of the 19th century, physics was at a crossroads. The experiments by Young had disconfirmed Newton's corpuscular theory of light and it was widely accepted that light is a wave. However, waves need a medium through which to propagate. Thus, if outer space is a vacuum, the question arose: How can light reach Earth from the stars? The widely accepted answer was that the universe is filled with a "luminiferous ether" transparent to ordinary matter but nevertheless sufficient to facilitate the propagation of light waves. Michelson and Morley [5] invented an interferometer that tested the existence of the luminiferous ether but obtained near null results. It is interesting that their results were not exactly null, and they collected so many data points that had they performed modern tests of statistical significance, they would have obtained statistical significance, arguably to the detriment of physics [6]. In most of the 20th century and the 21st century, it has been accepted that there is no luminiferous ether and that the theoretical effect size is zero. Thus, we come to the M & M problem pertaining to reproducibility: How reproducible are the Michelson and Morley experiments? On the one hand, physicists consider them highly reproducible; valid interferometers consistently produce near null results. On the other hand, according to the foregoing conceptions of what constitutes successful replications that imply a dependence on effect size, the Michelson and Morley experiments are not reproducible because the theoretical effect size is zero and so there is no way to correctly obtain, and replicate, statistically significant *p*-values. Clearly, there is something very wrong with the foregoing notions of reproducibility.

Trafimow [4] suggested a completely different approach. Rather than asking about characteristics relative to hypotheses, necessary for all *p*-value computations, it is alternatively possible to ask whether an original and replication study result in sample statistics that are close to the corresponding population parameters they are used to estimate. If sample statistics are close to corresponding population parameters in the original and replication studies, that counts as a successful replication, thereby solving the M & M problem. This approach emphasizes getting the empirical facts straight before drawing conclusions about their implications for substantive hypotheses. An additional advantage to the approach is that it solves the problem of effect size inflation demonstrated by the Open Science Collaboration [1] and discussed subsequently by many authors (e.g., [7–12] and [4]). As will be elaborated later, the present focus is on obtaining good sample estimates of population parameters—however large or small the population parameters may be—and not on obtaining sample effect sizes sufficiently large to render *p*-values under threshold.

Let us move directly to an important distinction between replication in an ideal universe, where the only differences between experiments are those engendered by random processes versus replication in the real universe where there are systematic differences too. Trafimow [4] denoted replication in the ideal universe as replication$_{ideal}$, and replication in the real universe as replication$_{real}$. As there are both systematic and random factors rendering difficult replication$_{real}$, whereas there are only random factors rendering difficult replication$_{ideal}$; the probability of replication$_{ideal}$ places an upper limit on the probability of replication$_{real}$. As most experiments in the social sciences are problematic even with respect to replication$_{ideal}$, they are even more problematic with respect to replication$_{real}$.

With the foregoing background in mind, let us proceed to what we see as the main limitation of Trafimow [4]. Specifically, very large sample sizes are needed to satisfy his equations. To understand this feasibility limitation, however, it is important to consider from whence Trafimow's equations came, which is the a priori procedure originally proposed by Trafimow [13], and expanded subsequently (e.g., [14–22]).

### 3. The A Priori Procedure (APP)

To understand the change in philosophy engendered by the APP, it is useful to consider an extremely simple example. Suppose that a researcher randomly and independently samples from a normally distributed population, and obtains the sample mean for a single group. The researcher hopes that the sample mean is close to the population mean. How many participants does the researcher need to collect to be confident that the sample mean is close to the population mean? The answer is to use Equation (1) below, where $n$ denotes the sample size, $f$ denotes the fraction of a standard deviation the researcher wishes to define as "close," and $z_{(1-c)/2}$ is the $z$-score that corresponds to the degree of confidence the researcher wishes to have of obtaining a sample mean within the specified degree of closeness ([23–25] and [13]):

$$n = \left( \frac{z_{(1-c)/2}}{f} \right)^2.$$ (1)

For example, suppose the researcher wishes to be 95% confident of obtaining a sample mean within 0.25 of the population mean. As the $z$-score that corresponds to the desire for 95% confidence is 1.96, the sample size needed is: $n = \left( \frac{1.96}{0.25} \right)^2 = 61.47$. Rounding upwards to the nearest whole number implies that the researcher needs 62 participants to meet specifications for precision $f = 25$ and confidence (95%). Although 62 participants might not seem particularly onerous, suppose that the researcher wishes to have much more precision at 95% confidence, so that $f = 0.1$. In that case, 385 participants are needed to meet specifications. Note that all computations can be carried out before obtaining any data; APP computations are pre-data.

The equations also can be used post-data. Trafimow and Myüz [14] analyzed journals in five areas of psychology to estimate precision, based on reported sample sizes, and found both a precision problem and that precision estimates across psychology areas differ dramatically. Trafimow et al. [15] provided a similar demonstration in the marketing field.

As the experiment becomes increasingly complex, two factors increase the need for more participants. As there are more groups, and so the researcher needs to obtain more sample means as estimates of corresponding population means, more participants are needed per group to ensure that all means in the experiment meet specifications. In addition, the total sample size is a function not only of the sample size per group, but also of the number of groups, so increasing the number of groups importantly increases the total sample size.

However, thus far the discussion in this section has focused on precision, confidence, and the number of groups. If one wishes to replicate, according to the conception of replication$_{\text{ideal}}$ explained in the previous section, the sample means must meet specifications not just in one experiment, but in two experiments. If one uses an appropriate APP equation to obtain the probability of obtaining sample means with the desired precision in one experiment, it is necessary to square that probability to obtain the probability of replication$_{\text{ideal}}$. Or considering the necessary sample size for replication, yet additional participants are needed to enable the experimenter to meet precision and confidence specifications for the $k$ groups in both the original and replication experiments. To dramatize the problem, Table 1 from Trafimow [1] shows that to have a 90% probability of replication, given precision and confidence specifications of 0.1 and 95%, respectively; when there are four groups; the researcher needs a total sample size of 2464. Of course, the researcher could insist on less stringent criteria for probability of replication, precision, or confidence; or even could conduct a simpler experiment (such as having two groups instead of four groups); but many of the values in Trafimow's Table 1 are not feasible for most researchers even so. What can be done?

### 4. Embracing Skewness

In a recent article, Trafimow et al. [16] considered the family of skew-normal distributions, of which the family of normal distributions is a subset (see [26–28] for a thorough

discussion). Normal distributions have two parameters: mean $\mu$ and standard deviation $\sigma$. In contrast, skew-normal distributions have three parameters.

**Definition 1.** *A random variable X is said to be a skew-normal random variable with location parameter $\xi$, scale parameter $\omega$ and shape parameter $\lambda$ if its probability density function is*

$$f(x) = 2\phi\left(x; \xi, \omega^2\right)\Phi\left(\lambda\frac{x-\xi}{\omega}\right), \tag{2}$$

*where $\phi\left(x; \xi, \omega^2\right)$ is the probability density function of a normal random variable with mean $\xi$ and variance $\omega^2$, and $\Phi\left(\lambda\frac{x-\xi}{\omega}\right)$ is the cumulative probability of the standard normal random variable at $\lambda\frac{x-\xi}{\omega}$. The mean, the variance, and the third moment of X are listed below.*

$$\mathrm{E}(X) \equiv \mu = \xi + \sqrt{\tfrac{2}{\pi}}\delta\omega, \ \ \mathrm{V}(X) \equiv \sigma^2 = \omega^2\left(1 - \tfrac{2}{\pi}\delta^2\right)$$
$$\text{and} \tag{3}$$
$$\mathrm{E}\left(X^3\right) = \xi^3 + 3\sqrt{\tfrac{2}{\pi}}\delta\omega\xi^2 + 3\omega^2\xi + \sqrt{\tfrac{2}{\pi}}\left(3 - \delta^2\right)\delta\omega^3,$$

*where $\delta = \frac{\lambda}{\sqrt{1+\lambda^2}}$.*

The mean $\mu$ is replaced by the location $\xi$, the standard deviation $\sigma$ is replaced by the scale $\omega$, and there is a shape parameter $\lambda$. When the shape parameter is zero, the distribution is normal, the mean and location are the same, and the standard deviation and location are the same; but when the shape parameter is not equal to zero, the mean and location differ, and the standard deviation and scale differ. In symbols, when $\lambda = 0$, $\xi = \mu$ and $\omega = \sigma$; but when $\lambda \neq 0$, $\xi \neq \mu$, and $\omega \neq \sigma$.

As the location is a parameter of all skew-normal distributions whereas the mean is a parameter only for normal distributions, the location is the more generally applicable parameter. Similarly, the scale is more generally applicable than the standard deviation. To expand the APP so that researchers are not forced into assuming normal distributions, Trafimow et al. [17] developed equations to compute the sample size needed to meet specifications for precision and confidence involving sample locations, as opposed to sample means, in the context of the family of skew-normal distributions. Given the often-made recommendations for researchers to perform data transformations to reduce skewness, one might predict that sample sizes would have to increase to meet specifications for precision and confidence, with respect to locations, in the presence of skewness. In contrast, the surprising finding was that skewness dramatically reduced the sample sizes needed to meet specifications; the more the skewness, the smaller the necessary sample size. Intuitively, the reason for this surprising effect is that skew normal distributions are taller and narrower than normal distributions, thereby increasing the probability of sampling from the bulk of the distribution rather than from a tail of the distribution. For example, suppose there is a single group and one wishes to estimate the sample size needed to reach precision of 0.10 at 95% confidence. The necessary sample size needed to meet the specifications is 385 when the shape parameter is 0, but it is only 146 when the shape parameter is 1. However, further skewness does not imply further strong effects, as even setting the shape parameter at 5 only reduces the sample size requirement to 138.

Well, then, we can now provide a practical solution to the issue of poor reproducibility in the soft sciences and the necessity for extremely large sample sizes that are not feasible for most researchers to obtain. As skewness decreases sample size requirements necessary to meet specifications for precision and confidence, all else being equal, it also should decrease sample size requirements for replication$_{\text{ideal}}$. In addition, because few distributions are normal whereas most distributions are skewed [29–31], most researchers ought to be able to avail themselves of the savings.

### 5. Three Demonstrations

It is convenient to fix the probability of replication at a single level and show how skewness influences sample sizes needed to reach that level. Remaining with the convention of 95% confidence, the probability of replication$_{ideal}$ is $0.95 \times 0.95 = 0.9025 \approx 90\%$. Thus, 90% will be used as our arbitrary replication$_{ideal}$ threshold in each of the following subsections. To reiterate, this is an upper bound for the probability of replication$_{real}$. The following subsections provide demonstrations of the benevolent effects of skewness on replication$_{ideal}$, in experiments with one sample, matched samples, or independent samples, respectively.

### 5.1. One Sample

We saw earlier that to have a 95% chance of obtaining a sample mean within one-tenth of a standard deviation of the population mean, it is necessary to have 385 participants. To address skewness, as explained earlier, it is necessary to use the more general location $\xi$ as opposed to the mean $\mu$, though the two are the same when the shape parameter is 0 (normal distribution). Well, then, suppose a mild degree of skewness, say, that the shape parameter is 0.5. Nonetheless, the necessary sample size to meet the same specifications and the same (90%) probability of replication$_{ideal}$ for the location is 158 (see Trafimow et al. [17], for mathematical derivations, tables and figures, and computer simulations). Subtracting 158 from 385 results in a difference of 227 and a savings of 59%. Of course, if the researcher is satisfied with less precision, both sample sizes and savings decrease. Figure 1 illustrates how the advantage conferred on the researcher by skewness, relative to normality, increases as the desired precision increases reading from right to left along the horizontal axis (small numbers indicate more precision). We used the method by Trafimow [13] to obtain the curve in Figure 1 representing normality and we used the method by Trafimow et al. [17] to obtain the curves in Figure 1 representing skew-normality.

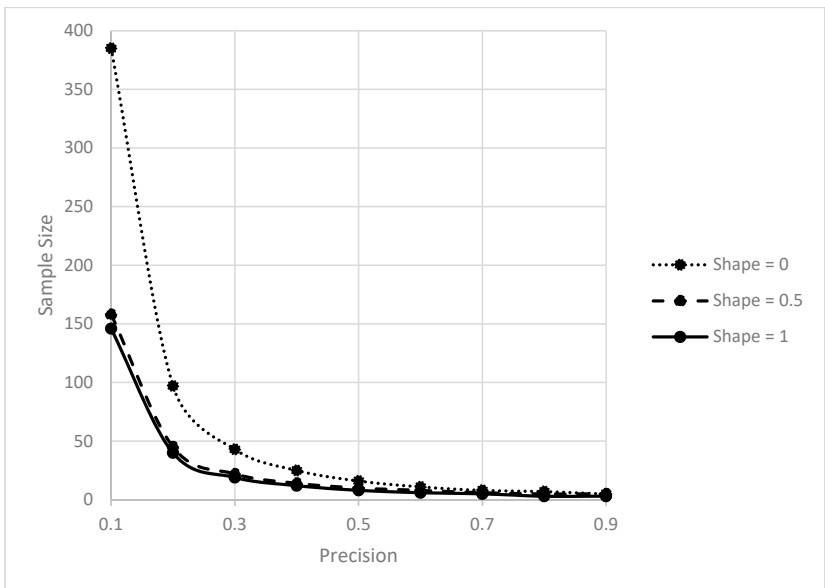

**Figure 1.** The sample size necessary for having a 90% probability of replication for estimating a single location parameter is expressed along the vertical axis as a function of the specification for precision along the horizontal axis and assuming a normal distribution (dotted curve), skew-normal distribution with shape = 0.5 (dashed curve), or a skew-normal distribution with shape = 1 (solid curve).

In addition, to show that most of the benefit of skewness occurs at very low levels of skewness, Figure 1 also shows how the sample size is influenced by the desired precision when the shape parameter is 1 rather than 0.5. Note that the solid curve (shape = 1) almost

overlaps the dashed curve (shape = 0.5), thereby indicating that most of the benefit of skewness occurs at the low level whereby the shape parameter is 0.5. This similarity can be considered an advantage for researchers because very little skewness is required to realize almost all skewness gains.

Although we used a 90% replication rate, this was for the twin reasons of conventionality and familiarity; not because of necessity. We thank a reviewer for suggesting that we consider a 95% replication rate which requires that confidence needs to be set at 0.975. In turn, the implication is that the required sample sizes for precision at the 0.10 level, when the shape parameter is set at either 0 or 0.50, are 503 and 212, respectively. These are more stringent than the corresponding samples sizes under a 90% replication rate (385 and 158, respectively). It is interesting that insisting on a more stringent replication rate of 95%, as opposed to 90%, increases the beneficial effect of skewness; the savings is now 291, rather than 227 as we saw earlier.

### 5.2. Differences in Matched Samples

Imagine a social psychology experiment where participants give their attitudes pertaining to a minority and majority target person. The researcher's goal is to obtain a precise estimate of the difference between attitudes towards the two target persons. How many participants does the researcher need to collect to have a 90% probability of replication$_{ideal}$?

Figure 2 illustrates the answer. Although Figure 2 resembles Figure 1, the numbers are slightly higher because we assumed a known standard deviation for Figure 1 and an unknown standard deviation for Figure 2. For example, assuming normality, the sample size needed for a 90% probability of replication$_{ideal}$ when precision is set at 0.1 is 387 rather than 385. In contrast, if the difference scores form a slightly skewed distribution (shape parameter equals 0.5), only 158 participants are needed. Thus, we again realize substantial savings by embracing skewness, even slight skewness. Additionally, again, the curves for shape parameters of 0.5 or 1 almost overlap, replicating the previous demonstration that slight skewness suffices for almost all skewness gains. (We used the procedure from Trafimow et al. [19], for the curve in Figure 2 representing normality and the procedure from Wang et al. [22], for the curves in Figure 2 representing skew-normality.)

As in the single sample case, it is possible to insist on a 95% replication rate instead of a 90% replication rate, keeping the precision level at 0.10. The implication is that the required sample sizes, when the shape parameter is set at either 0 or 0.50 for both matched groups, are 504 and 216, respectively. This contrasts with the corresponding values of 387 and 158 under a 90% replication rate. Thus, the skewness benefit of 288, under a 95% replication rate, is greater than the skewness benefit of 229, under a 90% replication rate.

### 5.3. Independent Samples

Imagine a typical experiment comparing two treatments for a disorder using two independent samples. The researcher wishes to have a precise and replicable estimate of the difference in locations between the two distributions. For the sake of simplicity, let us assume that the researcher has equal sample sizes and both distributions have the same shape (both shape parameters = 0, 0.5, or 1.0). (It is not necessary to make these simplifying assumptions. The equations by Trafimow et al. [19], and by Wang et al. [22], handle different sample sizes and different shapes, respectively.) Figure 3 illustrates the sample sizes needed for a 90% replication probability at various precision levels. As Figure 3 shows, when excellent precision ($f = 1$) is demanded, the necessary sample size, per group, is quite large when the distribution is normal (770), decreases to 480 when the shape parameter is 0.5, and decreases slightly more to 479 when the shape parameter is 1. Sample size requirements and differences between the curves are less dramatic, as usual, when less precision is demanded. Again, we see an impressive decrease in the sample size requirement in the presence even of slight skewness (e.g., $770 - 479 = 321$). (We used a procedure from Trafimow et al. [19], for the curve in Figure 2 representing normality and a procedure from Wang et al. [22] for the curves in Figure 2 representing skew-normality.)

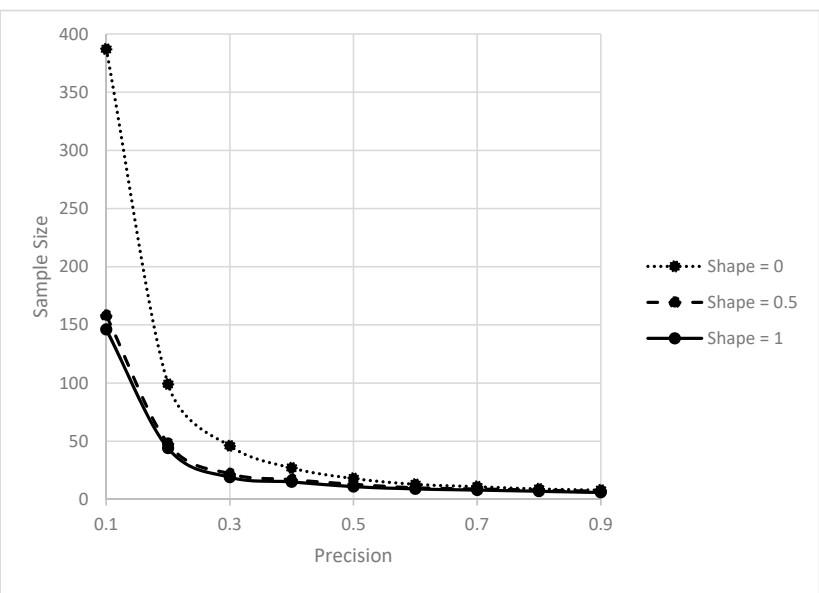

**Figure 2.** The sample size necessary for having a 90% probability of replication for estimating a difference in locations in two matched samples is expressed along the vertical axis as a function of the specification for precision along the horizontal axis and assuming a normal distribution (dotted curve), a skew-normal distribution with shape = 0.5 (dashed curve), or a skew-normal distribution with shape = 1 (solid curve).

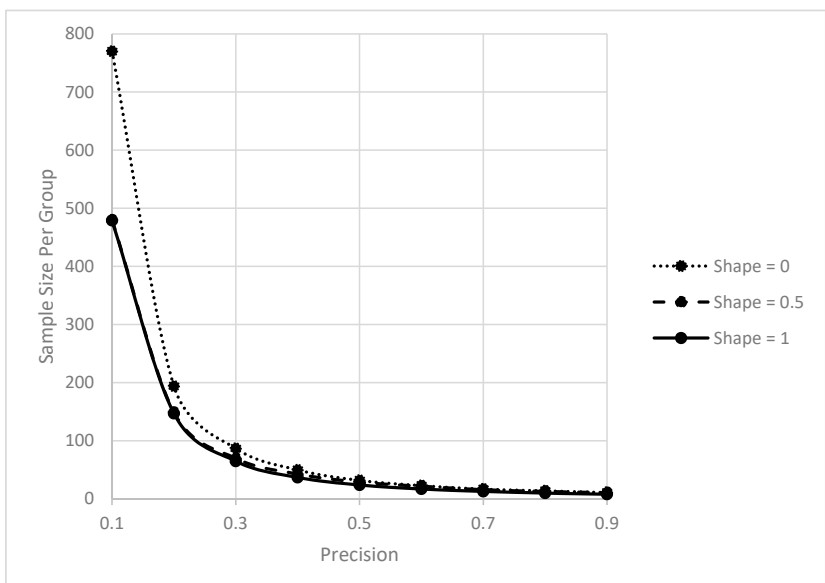

**Figure 3.** The sample size per group necessary for having a 90% probability of replication for estimating a difference in locations in two equally sized independent samples is expressed along the vertical axis as a function of the specification for precision along the horizontal axis and assuming a normal distribution for both samples (dotted curve), a skew-normal distribution with shape = 0.5 for both samples (dashed curve), or a skew-normal distribution with shape = 1 for both samples (solid curve).

However, Figure 3 underestimates what might be considered the true sample size savings illustrated in Figure 4. Consider that when there are independent samples, the total sample size is the sum of the two group sample sizes. Returning to our example, if the distributions are normal, the total sample size is twice the value mentioned earlier: it is 1540. In contrast, if the shape equal 0.5, the total sample size requirement is 958. Thus, slight skewness enables a savings of 1540 − 958 = 582. As usual, both the sample size requirement, and the savings, are reduced if less precision is demanded.

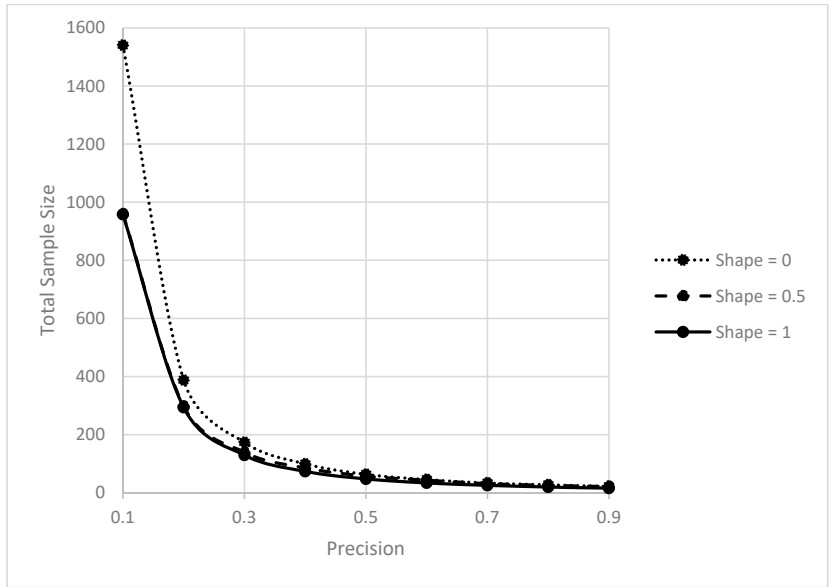

**Figure 4.** The total sample size necessary for having a 90% probability of replication for estimating a difference in locations in two equally sized independent samples is expressed along the vertical axis as a function of the specification for precision along the horizontal axis and assuming a normal distribution for both samples (dotted curve), a skew-normal distribution with shape = 0.5 for both samples (dashed curve), or a skew-normal distribution with shape = 1 for both samples (solid curve).

## 6. Discussion

The foregoing demonstrations render obvious the sample size savings researchers can enjoy merely by embracing skewness, featuring locations rather than means (and scales rather than standard deviations). However, are the calculations feasible? To see that they are, let us consider the population level relations between means and locations and between standard deviations and scales, and the sample relations too.

Equation (4) provides the relations at the population level by Equation (3),

$$\xi = \mu - \sqrt{\frac{2}{\pi}}\delta\omega \text{ and } \omega^2 = \frac{\sigma^2}{1 - \frac{2}{\pi}\delta^2}, \tag{4}$$

where $\delta = \frac{\lambda}{\sqrt{1+\lambda^2}}$.

Recognizing that researchers rarely or never have access to population parameters, these usually must be estimated. Practically all statistical packages provide sample statistics such as the mean $\overline{X}$, standard deviation $S$, and skewness $\hat{\gamma}_1$. These sample statistics can be made to render estimates of population shape, location, and scale parameters.

It is useful to commence by obtaining an estimate of delta $\hat{\delta}$ using Equation (5).

$$|\hat{\delta}| = \sqrt{\frac{\pi}{2} \frac{\hat{\gamma}_1^{\frac{2}{3}}}{\hat{\gamma}_1^{\frac{2}{3}} + \left(\frac{4-\pi}{2}\right)^{\frac{2}{3}}}}, \tag{5}$$

where the sign of $\hat{\delta}$ is the same as the sign of $\hat{\gamma}_1$.

In turn, it is easy to obtain an estimate of the shape or skewness parameter $\hat{\lambda}$ using Equation (6):

$$\hat{\lambda} = \frac{\hat{\delta}}{\sqrt{1 - \hat{\delta}^2}}, \tag{6}$$

Rewriting Equation (4) in terms of sample estimates, as opposed to population parameters, renders Equation (7):

$$\hat{\xi} = \overline{X} - \sqrt{\frac{2}{\pi}}\hat{\delta}\hat{\omega} \text{ and } \hat{\omega}^2 = \frac{\hat{\sigma}^2}{1 - \frac{2}{\pi}\hat{\delta}^2}. \tag{7}$$

Thus, it is feasible for researchers to replace mean and standard deviation with location and scale, respectively, and thereby benefit from sample size savings such as those demonstrated earlier. Even without sample size savings, the proposed replacement would be desirable because location and scale are more generally applicable than mean and standard deviation.

Although the proposed solution to irreproducibility is both practical and without monetary cost, there is an important limitation, and we wish to be upfront about it. Specifically, despite the generally greater applicability of the family of skew-normal distributions over the family of normal distributions, researchers should not take this as indicating that the family of skew-normal distributions is always applicable. For example, a distribution might be bimodal, in which case estimating locations and scales may not be a good strategy. Consequently, although the present recommendation will work much of the time—whenever the family of skew-normal distributions is applicable—there is no substitute for researchers paying careful attention to the distributions they feel they are likely to obtain, if planning a study; or in paying careful attention to the distributions they actually get, for data already collected.

The present contribution can be summarized easily. Trafimow [1] did much of the work by suggesting a better way to think about reproducibility than previous ways involving $p$-value thresholds. Specifically, Trafimow's notion of successful replications pertains to original and replication experiments resulting in sample statistics close to corresponding population parameters. Trafimow and colleagues also provided relevant APP equations assuming normal or skew-normal distributions [13,15–22]. However, the major limitation of Trafimow [1] is that he employed the normal equations because the skew-normal equations had not been invented yet. The consequence is that the necessary sample sizes for reasonable criteria for replication probabilities and precision are too onerous to expect most researchers to obtain them. Thus, there is an important practical problem. The present work addresses that problem by showing how embracing skewness, which is usually present anyway in the data that researchers obtain, importantly mitigates the practical problem. Additionally, it does so even at very slight levels of skewness (e.g., shape parameter equals 0.5). If researchers would routinely estimate location and scale parameters instead of, or in addition to, mean and standard deviation parameters; replication probabilities at currently used sample sizes would increase dramatically and we present a practical demonstration, using data from a recently published article, in the Appendix A. The obvious benefits to the soft sciences of better replication probabilities can be realized without the necessity of acquiring any additional resources, merely by following the simple and feasible prescriptions described here.

**Author Contributions:** Methodology, T.W. and C.W.; Writing—original draft, D.T. All authors have read and agreed to the published version of the manuscript.

**Funding:** This research received no external funding.

**Institutional Review Board Statement:** Not applicable.

**Informed Consent Statement:** Not applicable.

**Data Availability Statement:** Not applicable.

**Conflicts of Interest:** The authors declare no conflict of interest.

## Appendix A

Consider a practical application of our proposal in the context of an experiment performed by Dolinska et al. [32] in *Basic and Applied Social Psychology*. They performed a well-conducted study using six experimental conditions, and a control condition for comparison purposes, for a total of seven conditions. Unlike typical research in psychology, these researchers reported estimates not just of normal parameters for each condition (mean and standard deviation), but of skew normal parameters too (location, scale, and shape). Although Dolinska et al. did not perform the a priori procedure before data collection, it is interesting to imagine that they had, and ask about the necessary sample sizes to meet criteria for precision and confidence (or replication rate). Suppose Dolinska et al. had wished to meet a 0.30 level for precision at 95% confidence (thereby implying a 90% replication rate). In that case, for the goal of comparing each experimental condition mean against the control condition mean, the necessary minimum sample size would be 87 under normality, a condition which they did not meet. Alternatively, we might suppose that Dolinska et al. had wished to meet the same criteria but for the goal of comparing each experimental condition location against the control condition location. In that case, assuming a small amount of skewness, so the shape parameters both equal 0.50, the necessary sample size for each condition would be 70, which Dolinska et al. barely met. By taking advantage of skewness, and using locations instead of means, Dolinska et al. met a replication criterion that they otherwise would have been unable to meet.

The foregoing implies consequences for their reported mean comparisons and location comparisons. Regarding mean comparisons, keeping the sample size at the published level, implies that confidence is 92% and so the replication rate is 85% for each comparison. In contrast, regarding locations, assuming a shape parameter of 0.50 or more in each of the conditions (supported by their data) implies a confidence level of 95% and a replication rate of 90%. As a replication rate of 90% is superior to a replication rate of 85%, we have a demonstration using published data, of the benefits of embracing skewness to increase reproducibility.

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
