# Peer review of "A Viable Approach to Mitigating Irreproducibility"

_stats, doi:10.3390/stats4010015_

Round 1

Reviewer 1 Report

The authors discussed a procedure to calculate the probability of replication using a skew distribution. They advocate that using a model with the presence of skewness decrease the necessary number of participants that are needed to ensure different levels of replicability. The topic is interesting and I believe that can be considered for publication in the Stats. I have some comments that should be incorporated:

1) The authors argued that “In contrast, the surprising finding was that skewness dramatically reduced 168 the sample sizes needed to meet specifications; the more the skewness, the smaller the 169 necessary sample size.”
In fact, this is surprising, intuitively I believed that the presence of higher skewness would imply the need for additional participants to capture the information from the tail. While the authors have cited previous work published by themselves, it would be nice to include a better discussion regarding this topic. In fact, I had to go through many papers published by the authors to follow the manuscript. Hence, please provide additional information in the sense that the readers can obtain the main ideas and understand the current paper.

2) Please consider the reproducibility of 95% instead of 90%. At least for me, this is a desirable value. If you can, please compare the difference of samples between both values ( 95% and 90%).

3) The authors argued that usually the population parameters from the skew-normal distribution is unknown, which is true, and suggested to use the moment estimators to obtain estimates values. However, the moment estimators are known to be highly biased and less efficient than the maximum likelihood estimates (MLE). Therefore, It would be nice to discuss the implications of using the MLE instead of moments estimators.

Author Response

The reviewer made three suggestions and we felt that all three were well taken.

First, the reviewer mentioned having had to go through previous papers to understand about how skewness decreases sample sizes needed to meet specifications for precision and confidence. We very much appreciate that the reviewer took the trouble to do this and we agree that more description could make life easier for the reader. Therefore, we added the extra discussion that the reviewer requested with a specific example for clarification.

Second, the reviewer suggested we try out a replication rate of 95% to compare with 90%. We did this in the revision. As the reviewer doubtless anticipated, a more stringent replication criterion demands more participants.

Third, the reviewer suggested including a discussion of maximum likelihood estimates compared to moment estimators. We agreed with the reviewer about the desirability of that discussion but did not want to interrupt the narrative flow. Consequently, we included that discussion by lengthening Footnote 10 to (a) explain what the reviewer said about bias and efficiency and (b) to suggest that moment estimators nevertheless provide a good starting point. Footnote 10 also provides a reference for a discussion of different ways to estimate skewness for those readers who want more information about skewness estimation.

Reviewer 2 Report

1. The results explained in p. 3, rows 210-214 are not new. Only the notations are new. It is taught also in uonversities in BG in bachelor courses in Statistics and it is inapropriate to site Trafimow(2017) for this. Such exaples as the one after these rows is appropriate for some course in statistics for psycologists, but not for a journal with name Stats.

2. P. 4, rows 176-178 According to me the sentence "Because skewness decreases sample size requirements necessary to meet specifications for precision and confidence, all else being equal, it also should decrease sample size requirements for replication_{ideal}" is not correct. In order to clarify this please explain how the Central Limit Theorem corresponds to this result. The last means that the results of the paper cannot be correct.

Author Response

The reviewer made two points. First, the reviewer was unhappy with a citation and so we added two additional citations.

Second, the reviewer felt that the Central Limit Theorem might be a problem for our argument. We disagree with this because the Central Limit Theorem applies to means, not to locations, and it is locations that are crucial for our argument. In addition, please see math that is attached but that does not come through when we cut and paste in this box. 

Reviewer 3 Report

In this paper a new approach to increasing reproducibility is proposed, by using the family of skew-normal distributions. It allows decreasing the sample size requirements for replication ideal as well as meeting the requirements for precision and confidence.

The paper contains interesting and relevant findings, with applicative potential to problems with less than optimal sample sizes available. Therefore, I consider it can be published, by taking into account the following suggestion: as the results presented could be used to address practical problems, please include, if possible, in the Appendix, some relevant results regarding the family of skew-normal distributions.

Author Response

The reviewer had only one suggestion, but it was really excellent. Specifically, this reviewer suggested we provide an Appendix, with a practical demonstration of the benefits of our proposal. We strongly agreed and provided it at the end of the manuscript.

Reviewer 4 Report

The question dealt with by the authors is of general interest. However, the material is presented in an unfriendly manner to readers. While formula (1) and its probabilistic background are easy to understand by every reader, it remains largely unclear which probabilistic statement is to be circumscribed by formula (2). Basic probabilistic background is not even mentioned by the authors. Mayor revision is necessary. It must be clear how large the amount of new probabilistic and statistical information is.

Author Response

We thank the reviewer for the comment. We agree with the reviewer and made changes accordingly. First, we included more equations describing the different moments leading up to the formula. Because of that, the formula is now number 4 rather than number 2. Second, the reviewer’s comment aided us in another way because we realized a typo in the equation. We have fixed it.

Round 2

Reviewer 1 Report

The authors have included most of my suggestions. I believe that the manuscript can be accepted in this current form.

Author Response

We thank the reviewer (a) for going above and beyond the call of duty in finding articles while reviewing the previous version, (b) for making suggestions that improved the manuscript, and (c) for supporting acceptance of the revision. More generally, we appreciate the reviewer's careful reading with respect to our manuscript. 

Reviewer 2 Report

Dear Authors, 

the asymmetry is usually managed by finite dimensional distribution of the considered characteristic. You can also replace the mean with the median and the tests that you use with Kruskal-Wallis confidence intervals. With other words according to me your problems come from the fact that you use methods which are usually applied for normal data or large sample size. 

Therefore I can not support such a publication. 

Author Response

The reviewer's comment is not appropriate to our asymmetry assumptions for the following reasons. First, the location of a skew normal distribution is not the same thing as the median and the replacement the reviewer suggests will not work.  Secondly, The Kruskal–Wallis test or one-way ANOVA on ranks (Kruskal-Wallis confidence interval) is a non-parametric method for testing whether samples originate from the same distribution, which use rankings of observations not observations themselves.  It will not provide an a priori interval because the former is based on rankings from obtained data whereas the latter is not (this is in addition to the problem that the mean is not the location if the skewness is not zero). Thirdly, in contradiction to what the reviewer said, we are not using methods usually applied for normal data; we do not wish to assume normality, which is why we used skew normal distributions. We wish the reviewer had read the manuscript more carefully, as the other two reviewers did, both of whom recommended publication.

Reviewer 4 Report

Unfortunately, the material of the present work is still presented in an unclear manner. It still remains largely undiscussed which probabilistic statement is statistically illustrated by the present work. Basic probabilistic background is still not even mentioned by the authors. In the revision, it did not become clear how large the amount of new probabilistic and statistical information is. For these reasons I suggest to reject this manuscript.

Author Response

We added some material pertaining to Equation 3 that we believe renders it clear. In turn, Equation 4 follows directly from Equation 3.